# OpenReview forum: "Do Large Language Models Know What They Are Capable Of?"
_ICLR.cc/2026/Conference — ICLR 2026 Poster_

### Official Review · Reviewer_GMRd · 2025-10-27

**Soundness:** 3
**Presentation:** 3
**Contribution:** 2
**Rating:** 4
**Confidence:** 3

**Summary:**

This paper investigates the self-awareness of large language models (LLMs), defined as their ability to predict their own success on a given task. The authors conduct three experiments to evaluate this capability: (1) eliciting in-advance confidence estimates on single-step tasks; (2) placing LLMs in a sequential resource-acquisition scenario where they must accept or decline "work contracts" based on their self-assessment, learning from in-context experience; and (3) tracking how confidence estimates evolve at each intermediate step on multi-step tasks.

The core findings are that: LLMs typically do not make more accurate confidence estimates; frontier LLMs successfully learn from past successes and failures  by increasing their risk aversion; and that reasoning LLMs are typically less accurate at predicting their success.

**Strengths:**

1. The paper tackles the issue of LLM self-awareness, which has significant implications. Understanding whether models "know what they don't know" is fundamental in many AI areas.

2. The evaluation is extensive, covering a wide range of modern LLMs from different families (Llama, GPT, Claude).

3. The paper confirms interesting LLMs behaviours, although not completely unexpected. For example,  in Experiment 2 that improved profitability stems from increased risk aversion

**Weaknesses:**

1. The central conclusion that LLMs are overconfident is not new, and is extensively documented in prior work, much of which is cited by the authors (e.g., Lin et al., 2022; Tian et al., 2023; Xiong et al., 2024). While the experimental setups are novel, the main takeaway is confirmatory rather than groundbreaking.

2. A significant opportunity seems to have been missed by not exploring methods to address the identified problem. The framework from Experiment 2, which measures expected profit, seems perfectly suited to use as a reward signal. Why did the authors not attempt to use this signal to improve model performance, for instance, through RL fine-tuning? A demonstration of even a simple mitigation strategy on a small-scale LLM would have substantially increased the paper's contribution.

**Questions:**

- The finding in Experiment 3 that reasoning-enabled models perform no better than their non-reasoning counterparts is surprising. Do the authors have a hypothesis for this? Could it be an artifact of the prompting strategy, the RLHF training, or does it suggest that current reasoning capabilities do not extend to robust self-assessment?

- The paper provides a strong foundation for future work. I would encourage the authors to build on this evaluation by exploring mitigation techniques, as this would be a highly valuable contribution.

---

> ### Author Response · Authors · 2025-11-24
>
> We are encouraged that __GMRd__ found the topic “fundamental” with “significant implications”. We also respectfully push back on the potential weaknesses raised by __GMRd__, and address their questions and other points below.
>
> ## Addressing Weaknesses:
>
> 1. **“The central conclusion that LLMs are overconfident is not new, and is extensively documented in prior work, much of which is cited by the authors (e.g., Lin et al., 2022; Tian et al., 2023; Xiong et al., 2024). While the experimental setups are novel, the main takeaway is confirmatory rather than groundbreaking.”**
>
>     - We respectfully disagree with the assertion that our main takeaway is confirmatory. While the result that LLMs are overconfident is indeed confirmatory, _this is not our central result_. Crucially, we study how overconfidence changes dynamically with in-context experience and how LLMs’ confidence relates to their decision making. We find significant differences among LLMs in how they update their confidence as they progress through Experiments 2 and 3. The prior work on LLM overconfidence in static settings is indeed a foundation that we build upon, but our experiments and results break new ground.
>
> 2. **“A significant opportunity seems to have been missed by not exploring methods to address the identified problem. The framework from Experiment 2, which measures expected profit, seems perfectly suited to use as a reward signal. Why did the authors not attempt to use this signal to improve model performance, for instance, through RL fine-tuning? A demonstration of even a simple mitigation strategy on a small-scale LLM would have substantially increased the paper's contribution.”**
>
>     - This is an excellent suggestion for future work and we agree that Experiment 2 could be adapted for an RL reward signal. However, _our goal for this work is to understand behaviors and trends in frontier LLMs, not to develop LLMs with better self-awareness of capability_. To provide more context for this goal: the authors have had discussions amongst ourselves that there may be certain settings in which it is not desirable for an LLM to have high self-awareness of capability for security reasons. It is not our aim with this paper to venture into those arguments. Rather, our aim with this work is only scientific investigation of LLMs, not engineering of LLMs.
>
> ## Responses to questions and other points:
>
> 1. **“The finding in Experiment 3 that reasoning-enabled models perform no better than their non-reasoning counterparts is surprising. Do the authors have a hypothesis for this? Could it be an artifact of the prompting strategy, the RLHF training, or does it suggest that current reasoning capabilities do not extend to robust self-assessment?”**
>
>     - Great question! We are surprised by this result as well, and while we have speculated about the reason for this result, we have not yet found a compelling hypothesis. Regarding the prompting strategy, we have aimed to make our prompts minimal, clear, and direct, so we would not expect that we are introducing artifacts through prompting. We do think it is likely that current reasoning capabilities do not extend to robust self-assessment, but we don’t have rigorous results to support this intuition. Relatedly, we are working to run Experiment 3 with GPT 5.1 and Sonnet 4.5, both with and without reasoning, and we are very curious to see how these results compare to our existing results. We hope to have these new results added to the paper within a week.
>
>
> 2. **With regard to risk aversion, which __GMRd__ commented on:**
>
>     - Firstly, we want to thank __GMRd__ and the other reviewers for their comments that led us to investigate risk aversion more carefully. In fact, our earlier claim about risk aversion increasing was incorrect. The corrected claim is that _high_ risk aversion (not an _increase in_ risk aversion) is what induces the observed behavior where a small decrease in overconfidence causes a large decrease in the contract acceptance rate. This is because when risk aversion is high, the decision threshold for accepting contracts (with reward=1, penalty=-1) is much greater than 50% estimated likelihood of success. Even if overconfidence drops only slightly, the estimated likelihood of success for many contracts can fall below this high decision threshold, resulting in a significantly reduced contract acceptance rate. In our updated paper, we quantitatively estimate risk aversion for each LLM, and clarify this mechanism.
>
>
> We thank __GMRd__ for these insightful comments which helped to improve our paper.

---

### Official Review · Reviewer_WUxE · 2025-10-31

**Soundness:** 3
**Presentation:** 3
**Contribution:** 3
**Rating:** 6
**Confidence:** 3

**Summary:**

The work is a quantitative analysis of the in-advance capability of a language model in determine success/failure outcome of a task. In particular, it studies whether the models can learn from recent successes/failures to make better go/no-go decisions when failure is costly, and update those predictions while working through multi-step, tool-using tasks.

The authors ran three experiments:
* Predicting the probability of success before attempting BigCodeBench Python coding tasks
* A modified setting where models sequentially attempt Python tasks in a “work contracts" setting that resembles a contextual one-armed bandit with an abstain option
* Stepwise confidence on multi-step SWE-Bench Verified, with a 70-tool-call budget

The work then presents several core findings: all models overestimate their success rates, they have better than random guesses, but their guesses have no trend of increasing discriminatory power with capability. Even with in-context learning, many models remain overconfident. Some frontier models improve profit in the bandit setting mainly by abstaining more, not by sharply improving discrimination/calibration. Finally, multi-step dynamics diverge between different models, with reasoning models not necessarily more calibrated than non-reasoning variants.

**Strengths:**

* The paper is very well written with great flow, good schematics, and no major grammatical or formatting errors
* The paper crystallizes an well-known phenomenon intuitively and anecdotally understood into a rigorous analysis
* Calibration is a very important problem to study and understand

**Weaknesses:**

* Unfortunately the world of foundation models move incredibly fast and the models tested in this work is already fairly dated. For example, GPT-4.1, Sonnet 3.5, etc. are no longer available. As noted by the authors, some of the behaviors characterized are divergent, and thus likely no longer relevant to the newest class of models (e.g. GPT-5).
* It would benefit the audience to draw connections between the in-advance setup of the work to calibration in traditional neural network architecture (e.g. [1], among others)

[1] Guo, C., Pleiss, G., Sun, Y., & Weinberger, K. Q. (2017, July). On calibration of modern neural networks. In International conference on machine learning (pp. 1321-1330). PMLR.

**Questions:**

I don't have more questions beyond what is mentioned in the weakness section.

---

> ### Author Response · Authors · 2025-11-24
>
> We are encouraged that __WUxE__ found the topic to be “very important”, that we had a “rigorous analysis”, and that our paper is “very well written with great flow”. __WUxE__ raised important and intriguing weaknesses, which have addressed:
>
> **Weakness 1: “The world of foundation models move incredibly fast and the models tested in this work is already fairly dated.”**
>
>   - Great point. We have now run our experiments on GPT 5.1 and Sonnet 4.5 in our analysis. For Experiments 1 and 2, the data from GPT 5.1 and Sonnet 4.5 is already included, and we expect results for Experiment 3 with these LLMs to be ready within a week (there were some delays due to changes in our access to compute resources in the time since our experiments were originally run). We also wish to emphasize that we intentionally include weaker LLMs because we wish to examine how our findings scale with model performance.
>
> **Weakness 2: “It would benefit the audience to draw connections between the in-advance setup of the work to calibration in traditional neural network architecture.”**
>
>   - We agree this is an intriguing point for discussion because, unlike LLMs, traditional NNs don’t provide verbalized confidence estimates. First, we note that in our Related Work section we indeed discussed prior work on token-level calibration of LLMs, which is neither “in-advance” nor “after-the-fact” and which is analogous to calibration in traditional NNs. Following __WUxE__’s suggestion, we have expanded this part of Related Works to discus calibration of non-LLM neural networks (lines 158-161).
>
> We thank __WUxE__ for raising these important points, which have led to important improvements in our paper.

---

### Official Review · Reviewer_DhWq · 2025-11-06

**Soundness:** 4
**Presentation:** 4
**Contribution:** 3
**Rating:** 8
**Confidence:** 4

**Summary:**

This paper investigates whether LLMs can predict whether (and how likely it is that) they can complete a task before performing it. This is tested both for single response cases using the BigCodeBench benchmark and in an agentic setup using the SWE-Bench verified benchmark. The authors find that all of the investigated models (recent GPT, Claude, and LLama models) overestimate the likelihood of completing a task successfully, and while some models can be steered through in-context learning to be more cautious about predicting that they can complete a task, this behaviour generally even persists in that scenario.

**Strengths:**

* While the research question is relatively simple, the paper evaluates the question thoroughly and considers the problem from multiple angles. And to the best of my knowledge, this paper introduces some new paradigms to evaluate model confidence.
* The paper is very well written, the experimental setup is very clear,  and it connects very well to previous work.
* The paper adequately discusses its limitations.

**Weaknesses:**

* While I think the task in Experiment 2 with fictional costs is an interesting approach, I was wondering whether this has been validated that LLMs can make such risk-based decisions when they have direct access to the underlying risks. In other words, is there evidence that LLMs can accurately compute the expected reward and base decisions on this implicit calculation? While I don't think this would change results here, since the confidence estimates still seem to be inaccurate anyways, it would be good to establish whether all LLMs can actually do this task since otherwise it may be challenging to derive anything meaningful from the model's decisions in this setup.

**Questions:**

See the weakness above.

---

> ### Author Response · Authors · 2025-11-24
>
> We are encouraged that __DhWq__ found that "the paper is very well written, the experimental setup is very clear, and it connects very well to previous work," and recognized how our work "introduced new paradigms to evaluate model confidence". We also thank the reviewer for their excellent question, which we discuss and investigate below:
>
> **“I was wondering whether this has been validated that LLMs can make such risk-based decisions when they have direct access to the underlying risks. In other words, is there evidence that LLMs can accurately compute the expected reward and base decisions on this implicit calculation?”**
>
> This question is subtle because the expected reward (i.e. expected utility, for a rational agent) from some outcome is typically not equal to the monetary payout of the outcome. For a rational risk averse agent obeying von Neumann-Morgenstern expected utility theory, payouts have decreasing marginal returns to utility. Hence, we address the question: _“Do LLMs make rational risk-based decisions when the underlying risks are known, suggesting that they compute expected reward according to a consistent utility function?”_ Prior work has studied this question, finding that LLMs are indeed fairly rational (generally more rational than humans) when making risk-based decisions when the underlying risks are known [1].
>
> In fact, we can also address a variant of this question with our data from Experiment 2. When underlying risks are unknown but an LLM has beliefs about the underlying risks, we can ask: _“Do LLMs make rational risk-based decisions according to their beliefs about the probabilities of events?”_  We answer precisely this question in our new Appendix (Appendix A), finding the answer to be generally yes.
>
> We thank __DhWq__ for raising this very interesting point, which, in combination with __VR9Y__'s comments, led us to write Appendix A. We think this is a substantial improvement in our paper.
>
> [1] Yiting Chen, Tracy Xiao Liu, You Shan, and Songfa Zhong. The emergence of economic rationality of GPT. Proceedings of the National Academy of Sciences, 120(51):e2316205120, 2023.

---

### Official Review · Reviewer_VR9Y · 2025-11-06

**Soundness:** 2
**Presentation:** 2
**Contribution:** 1
**Rating:** 2
**Confidence:** 3

**Summary:**

This work investigates whether LLMs know how likely they will succeed in a given task before attempting the task. The authors conducted three experiments in which LLMs performed confidence estimation in different settings and scenarios. Through the experiments, the study find that LLMs are generally overconfident to the given tasks, but still be able to identify tasks that they are capable of better-than-random. The work also shows that for many LLMs, such 'self-awareness' effect doesn't improve with models' increasing general ability, in-context experiences, and reasoning ability. Overall, the work reveals LLMs' poor self-awareness of capability, which may potentially hinder LLMs' application in high-stakes scenarios.

**Strengths:**

1. Comprehensive experimental design and clear visualization of the results.
2. Provide empirical evidence of 'self-awareness of ability' across different LLMs with different sizes.

**Weaknesses:**

1. Need an explicitly defined research gap, motivation, and why the study can fill the gap. The contributions of this work have not been highlighted from previous work.
2. How LLMs know what they are capable of is quantified by self-reported confidence in this study. Auto-regressive LLM models trained by next-token-prediction solve the confidence estimation given tasks in a fundamentally different manner from humans, making it sort of ambiguous to represent 'real confidence' in the tasks merely by reported scores. This could limit the implications of the study.

**Questions:**

1. What's the difference between in-advance/answer-free confidence estimation and retrospective estimation? Why use such ways of confidence estimation? Should the formers work better in this case?
2. How is actual success rate in Exp1 calculated? Doing each task for multiple times which produces a success rate for each individual task, or doing each task once that produces an overall success rate for the whole benchmark?
3. In Exp1, what is the best accuracy of predicting actual success rate with the reported confidence? And worth checking whether rephrasing the confidence estimation prompts can improve the prediction of actual success rate. For example, if asking LLMs to rate in-advance difficulty of each task, rather than the confidence, will the difficulty rating predict success rate well?
4. In Exp2, how many unique tasks are there in the pool of S and F tasks? Will the repetition of these tasks in the 9-contract sequences affect the results?

---

> ### Author Response · Authors · 2025-11-24
> **Comment 1 of 2**
>
> We are encouraged that __VR9Y__ found our experiments comprehensive and our results clear. We are especially grateful that __VR9Y__ **pointed out two important weaknesses, which we have now addressed.** We first respond to the weaknesses, then to the other questions.
>
> ## Addressing Weaknesses
>
> 1. **"Need an explicitly defined research gap, motivation, and why the study can fill the gap."**
>
>     - We believe our work fills a significant research gap though we agree that we did not define it explicitly enough. We have rewritten the first two paragraphs of the introduction to clarify this, including clarifying the significance of in-advance confidence. As discussed in our overall comment above, the research gap is the following: Prior work has not drawn the connection between LLM confidence and go/no-go decision making about whether to attempt a task, which is a key aspect of agent decision making. When attempting a task bears costs (as is typical for long tasks), well-calibrated in-advance confidence is essential for go/no-go decision making. After-the-fact confidence–which is the focus of most prior LLM calibration studies–is not relevant in this context because the go/no-go decision has already been made.
>
> 2. **"LLM models trained by next-token-prediction solve the confidence estimation given tasks in a fundamentally different manner from humans, making it sort of ambiguous to represent 'real confidence' in the tasks merely by reported scores."**
>
>     - This is an important and interesting point that we address by analyzing the consistency between LLM decision making and self-reported confidence. Specifically, we investigate whether LLMs’ decision making under risk is rational under the expectations defined by their self-reported confidence estimates. If their decision making is indeed rational under their self-reported confidence, we argue that this lends support to the hypothesis that their self-reported confidence is reflective of a “real” underlying confidence that determines their decision making. However, we acknowledge that this does not constitute a rigorous proof that LLMs’ self-reported confidence is equal to “real” underlying confidence, nor does it address the deeper philosophical question of what “real” confidence would mean for an LLM.
>
>     - Our analysis of this point is in a new appendix (Appendix A). We find that LLMs’ decision making is indeed approximately rational according to von Neumann-Morgenstern utility functions with expectations equal to the LLMs’ self-reported confidence estimates.
>
> ## Responses to VR9Y’s questions:
>
> 1(a). **"What's the difference between in-advance/answer-free confidence estimation and retrospective estimation?”**
>
>   - In-advance confidence on a task is estimated before attempting the task, whereas after-the-fact (retrospective) confidence is estimated after the task is completed but before the outcome (success or failure) is known. In our revised paper, these terms are defined in the first and second paragraphs of the introduction.
>
> 1(b). **"Why use such ways of confidence estimation? Should the formers work better in this case?"**
>
>   - Great question! In-advance confidence is relevant when a go/no-go decision must be made before a task is attempted. We do not expect in-advance confidence estimates to be more accurate than after-the-fact estimates—the key point is that after-the-fact confidence estimates are not relevant when a go/no-go decision must be made before a task is attempted. A contract example of when in-advance confidence estimates are necessary is deciding whether to accept a work contract to perform work that one has not already performed (as is typical of work contracts).
>
> 2. **"How is actual success rate in Exp1 calculated? Doing each task for multiple times which produces a success rate for each individual task, or doing each task once that produces an overall success rate for the whole benchmark?"**
>
>   - The latter. Success rate is the probability that the LLM will succeed on a randomly-sampled task from the benchmark. This is now specified on line 217.
>
> **Continued in next comment**

---

> ### Author Response · Authors · 2025-11-24
> **Comment 2 of 2**
>
> **Continued from above**
>
> 3(a). **"In Exp1, what is the best accuracy of predicting actual success rate with the reported confidence?”**
>   - Good question—we have added a new plot (panel B of figure 1) to explicitly show prediction accuracy. Sonnet 4.5 has the best prediction accuracy.
>
> 3(b). **“And worth checking whether rephrasing the confidence estimation prompts can improve the prediction of actual success rate. For example, if asking LLMs to rate in-advance difficulty of each task, rather than the confidence, will the difficulty rating predict success rate well?"**
>   - Great question. We emphasize that we indeed expect that alternate prompts, such as prompts that explicitly direct the LLM to avoid overconfidence, would produce more accurate estimates. However, we aim to elicit estimates without giving the LLMs such “hints” because we wish to investigate their confidence without influencing their confidence. For this reason, we wrote prompts (which are stated in Appendix B) to be maximally clear and direct, rather than to induce the highest accuracy.
>     - It is an interesting idea to prompt the LLMs to rate task difficulty rather than confidence. We have run this experiment on GPT 4.1 and GPT 5.1, and we found that the resulting AUROC scores were very similar to, but slightly lower than, the AUROC scores from in-advance confidence estimates. Specifically, the AUROC scores are:
>         - GPT 4.1: From difficulty rating: 0.61. From confidence estimate: 0.62
>         - GPT 5.1: From difficulty rating: 0.58. From confidence estimate: 0.60
>
> 4. **"In Exp2, how many unique tasks are there in the pool of S and F tasks? Will the repetition of these tasks in the 9-contract sequences affect the results?"**
>   - There are 271 S tasks and 193 F tasks. Importantly, we sample without replacement when constructing the sequences, so there are no repeated tasks in any sequence. Thank you for clarifying these points. We have now added these points to Appendix C.
>
> We thank __VR9Y__ again, as their comments led to significant revisions which we feel have substantially improved the paper.

---

### Author Response · Authors · 2025-11-24
**Revised Paper and Response to Key Points**

We thank the reviewers for their helpful, insightful, and overall positive feedback. We are encouraged that they found the topic important (__WUxE__, __GMRd__), the presentation clear (__DhWq__, __WUxE__, __GMRd__), and our approach novel, including introducing "some new paradigms to evaluate model confidence" (__DhWq__). We highlight three key points below, and provide more detailed responses in our individual comments.

## 1. Research gap and motivation for studying in-advance confidence:

We are grateful to __VR9Y__ for pointing out that we had not clearly explained the research gap nor our motivation for studying in-advance confidence. We have now clarified these points (1st and 2nd paragraph of introduction), which we think substantially improves the paper.
To briefly describe the research gap: Prior work has not drawn the connection between LLM confidence and go/no-go decision making about whether to attempt a task, which is a key aspect of agent decision making. When attempting a task bears costs (as is typical for long tasks), well-calibrated in-advance confidence is essential for go/no-go decision making. After-the-fact confidence–which is the focus of most prior LLM calibration studies–is not relevant in this context because the go/no-go decision has already been made. (We thank __WUxE__ for introducing the "go/no-go" terminology).


## 2. On LLM’s self-reported confidence:

__VR9Y__ raised an important point about whether LLM's self-reported confidence estimates accurately reflect "real confidence". As we are interested in LLM's confidence to the extent that it informs their decision making, we address this point by asking the questions:

(1) Is an LLM's self-reported confidence an accurate predictor of its decision making?

(2) Are LLMs' decisions consistent with rational choice under their self-reported expectations?

Using the data from Experiment 2, we answer both questions in the affirmative, and we quantitatively estimate each LLM's von Neumann-Morgenstern utility function. This is described in a new Appendix section (Appendix A and Figure 5). While this does not prove that LLMs' self-reported confidence is faithful, we argue that it lends support to this hypothesis.


## 3. New Experiments: Evaluating the most recently-released LLMs:

__WUxE__ rightly pointed out that we had not tested the most recently-released LLMs. We have now run Experiments 1 and 2 (and are in the process of running Experiment 3) on GPT 5.1 and Sonnet 4.5. Hence, our experiments now cover the current frontier of the Llama, GPT, and Claude families. We also emphasize that we intentionally include a wide range of LLMs, including relatively less capable LLMs, because we are interested in trends in how self-awareness of capability scales with general performance.

Detailed responses to the points raised by each reviewer are included in the individual rebuttals below. We again thank all the reviewers for their questions and insights, which we believe have significantly improved the paper.

---

### Meta-Review · Area_Chair_V3Pf · 2026-01-05

**Summary:**

The authors study whether LLMs can predict their own likelihood of success before attempting a task, how such confidence evolves during multi-step agentic tasks, and whether in-context experience of failure improves go/no-go decisions when failure is costly. Reviewers broadly found the topic important and the experiments clear, but raised some concerns about (i) clarifying the research gap vs prior calibration work and the motivation for in-advance confidence, (ii) whether self-reported confidence meaningfully reflects an underlying “real” confidence, (iii) model coverage being dated given the pace of frontier models, and (iv) whether the contribution is mainly confirmatory (overconfidence is known) without mitigation. The rebuttal substantially strengthened the paper by clarifying the gap, adding a new analysis showing decisions are approximately rational given reported probabilities (utility estimation), expanding related work to connect to classic calibration, and adding new experiments on newer frontier models (GPT-5.1, Sonnet 4.5). Overall, remaining concerns are mostly about scope.

**Reviewer Concerns:**

Addressed:
Unclear research gap: clarified why in-advance confidence matters for go/no-go decisions under costs vs retrospective calibration.
Meaning of self-reported confidence: added analysis showing decisions are rational given reported probabilities.
Dated model set: added experiments on GPT-5.1 and Sonnet 4.5.
Connection to classic calibration: expanded related work to include traditional calibration.

Perhaps still open:
Primary takeaway partly confirmatory: I guess this cannot be easily changed and doesn't have to be fatal
No mitigation method demonstrated: paper focuses on measurement rather than improving calibration
Experiment 3 partially still running.

**Reviewer Scores:**

DhWq: 8 to 8: concern about rational risk-based decision making addressed; likely unchanged.
WUxE: 6 to 7: dated-model critique addressed with GPT-5.1/Sonnet 4.5; stronger positioning/related work.
GMRd: 4 to 5: still sees confirmatory + wants mitigation, but rebuttal clarifies novelty and adds utility/risk-aversion analysis.
VR9Y: 2 to 4: major issues they raised was the gap definition and confidence-faithfulness concerns, these were directly addressed

---

### Decision · Program_Chairs · 2026-01-26

Accept (Poster)